# Enhanced Cognition and Neurogenesis in miR-146b Deficient Mice

**DOI:** 10.3390/cells11132002

**Published:** 2022-06-22

**Authors:** Keerthana Chithanathan, Kelli Somelar, Monika Jürgenson, Tamara Žarkovskaja, Kapilraj Periyasamy, Ling Yan, Nathaniel Magilnick, Mark P. Boldin, Ana Rebane, Li Tian, Alexander Zharkovsky

**Affiliations:** 1Department of Physiology, Institute of Biomedicine and Translational Medicine, University of Tartu, Ravila 19, 50411 Tartu, Estonia; keerthana.chithanathan@ut.ee (K.C.); ling.yan@ut.ee (L.Y.); li.tian@ut.ee (L.T.); 2Department of Pharmacology, Institute of Biomedicine and Translational Medicine, University of Tartu, Ravila 19, 50411 Tartu, Estonia; kelli.somelar@ut.ee (K.S.); monika.jurgenson@ut.ee (M.J.); tamara.zarkovskaja@ut.ee (T.Ž.); 3Department of Biomedicine, Institute of Biomedicine and Translational Medicine, University of Tartu, Ravila 19, 50411 Tartu, Estonia; kapilraj.periyasamy@ut.ee (K.P.); ana.rebane@ut.ee (A.R.); 4Department of Molecular and Cellular Biology, Beckman Research Institute of City of Hope National Medical Center, Duarte, CA 91010, USA; n.magilnick@gmail.com (N.M.); mboldin@coh.org (M.P.B.)

**Keywords:** miR-146b, cognition, anxiety, astrocytes, microglia, neurogenesis, neuronal development, *Gdnf*

## Abstract

The miR-146 family consists of two microRNAs (miRNAs), miR-146a and miR-146b, which are both known to suppress a variety of immune responses. Here in this study, we show that miR-146b is abundantly expressed in neuronal cells, while miR-146a is mainly expressed in microglia and astroglia of adult mice. Accordingly, miR-146b deficient (*Mir146b*-/-) mice exhibited anxiety-like behaviors and enhanced cognition. Characterization of cellular composition of *Mir146b*-/- mice using flow cytometry revealed an increased number of neurons and a decreased abundancy of astroglia in the hippocampus and frontal cortex, whereas microglia abundancy remained unchanged. Immunohistochemistry showed a higher density of neurons in the frontal cortex of *Mir146b*-/- mice, enhanced hippocampal neurogenesis as evidenced by an increased proliferation, and survival of newly generated cells with enhanced maturation into neuronal phenotype. No microglial activation or signs of neuroinflammation were observed in *Mir146b*-/- mice. Further analysis demonstrated that miR-146b deficiency is associated with elevated expression of glial cell line-derived neurotrophic factor (*Gdnf*) mRNA in the hippocampus, which might be at least in part responsible for the observed neuronal expansion and the behavioral phenotype. This hypothesis is partially supported by the positive correlation between performance of mice in the object recognition test and *Gdnf* mRNA expression in *Mir146b*-/- mice. Together, these results show the distinct function of miR-146b in controlling behaviors and provide new insights in understanding cell-specific function of miR-146b in the neuronal and astroglial organization of the mouse brain.

## 1. Introduction

MicroRNAs (miRNAs) are a family of small, endogenous, noncoding RNAs which are approximately 22 nucleotides long and act at post-transcriptional level to regulate gene expression via binding to partially complementary mRNAs [1,2,3]. The brain expresses more distinct and a relatively high number of miRNAs than any other tissue in vertebrates [4]. Early microarray profiling analyses have shown that miRNAs are differentially expressed in brain regions [5] and in a cell-specific manner [6]. As miRNAs can influence expression of many target mRNAs, they are capable of modulating various physiological processes in the brain, such as neural differentiation [7] and synaptic plasticity [8], and are thereby proposed to regulate to regulate complex animal behaviors including cognition [9] and anxiety [10]. In addition, miRNAs are involved in the several brain pathologies including neurodegenerative and psychiatric disorders [11,12,13].

The miR-146b belongs to the miR-146 family consisting of two miRNAs, miR-146a and miR-146b (miR-146a/b), which are encoded by two distinct genes located on different chromosomes and regulated by different pathways. Accordingly, the expression of miR-146a is upregulated through the nuclear factor kappa B (NF-κB) pathway [14], while miR-146b expression has been shown to be activated by signal transducer and activator of transcription (STAT) proteins [15,16]. As miR-146a/b differ only in two nucleotides, they probably target the same or a very similar set of genes [17,18]. The well-known target genes for miR-146a/b are TNF receptor-associated factor 6 (TRAF6) and interleukin-1 receptor-associated kinase 1 (IRAK1), both of which are from the NF-κB pathway and have been characterized already by an initial study on miR-146a/b [19]. Accordingly, there is ample evidence concerning the participation of the miR146 family in negative regulation of immune responses [14,15,16,17,18,20]. 

In the central nervous system, miR-146a has been reported to have immune suppressive function in microglia [21,22]. In addition, miR-146a has been related to autism, neural lineage determination, neurite outgrowth [23], differentiation of neural stem cells and hippocampal-dependent memory impairments [24]. The functions of miR-146b in the nervous system have been characterized by some existing studies. For example, one study showed that overexpression of miR-146b inhibited inflammatory responses via suppression of the activation of the NF-κB signaling in the brain in the rat encephalopathy models [25]. Another study in rats has shown that miR-146b overexpression with lentivirus vector could inhibit the proliferation of primary hippocampal neural stem cells [26]. A recent study has identified miR-146b as a candidate modulator of microglial activation [27]. In addition, miR-146b has also been reported to be a prognostic biomarker of gliomas, as it inhibits glioma cell proliferation and migration and induces apoptosis [28,29,30,31]. 

In this study, we used miR-146b deficient and wild type mice to explore the role of miR-146b in brain functions. We detected high expression of miR-146b in neuronal cells of the brain and observed that loss of miR-146b facilitates learning abilities and induces anxiety-like behaviors. Next, we found that lack of miR-146b affects cellular composition of the brain, reflected by the higher number of neurons, reduced number of astrocytes and increased hippocampal neurogenesis. In addition, we detected that miR-146b target GDNF mRNA is upregulated in *Mir146b*-/- mice in the hippocampus. No microglial activation or signs of neuroinflammation were observed in *Mir146b*-/- mice. Together, these results highlight the function of miR-146b in controlling cellular organization and behaviors in mice.

## 2. Materials and Methods

### 2.1. Animals and Experimental Design

*Mir146b*-/- mouse line on C57BL/6J background was generated by deletion of miR-146b encoding gene from mouse chromosome 19 as previously described in [17]. *Mir146b*-/- and corresponding *WT* mice used for this study were obtained by crossing *Mir146b+/-* heterozygous mice maintained and bred in the animal facility at the Laboratory Animal Centre at the Institute of Biomedicine and Translational Medicine, University of Tartu, according to the Institute’s regulations. The generated animals were genotyped using the primer sequence:

146b locus 5′ forward primer- 5′ CTCACACTCTTGTTCTTACCCAGTTCTT 3′; 146b locus 3′ reverse primer- 5′ CAAACAAACAAACAAAAGGTTCAGCTAAG 3′; 146b locus internal reverse primer-5′ACACACAGGGCATATGAGATCAGTTGGTT 3′ and same generation littermates were used in experiments. Two–three months old male mice were used for all experiments, which were undertaken in agreement with the guidelines established in the principles of laboratory animal care (Directive 2010/63/EU). All the mice were group-housed with a 12-h light/dark cycle with food and water available ad libitum. The Animal Experimentation Committee at the Estonian Ministry of Agriculture (no. 183, 2021) approved the experimental protocol. The sequence of the experiments performed is summarized in Table 1. The age of animals at the start of the experiments was approximately 2 months old. The age of animals at the time of sacrifice was approximately 3 months old. 

### 2.2. Novel Object Recognition Test (NORT)

NORT was performed as described by [32] in open chamber 50 cm × 50 cm × 50 cm (L × W × H), made up of brown wood. The objects were opaque glass cups of similar textures and colors but different sizes and shapes, and were heavy enough to prevent the mice from moving them. The experiment consisted of three phases: habituation, training and retention. During habituation phase, the animals were allowed to explore the empty arena without presence of any object for 5 min. Twenty-four hours later, in training phase, two identical objects were placed on a diagonal (both 10 cm from the corner) and each mouse was allowed to explore in the field for 5 min. The amount of time each mouse spent exploring both objects was recorded. Either 2 or 24 h later (retention phase), the mice explored the arena with presence of one familiar object and one novel object to measure their short-term recognition memory (STM) and long-term recognition memory (LTM), respectively. A preference ratio for each mouse was expressed as percentage of time spent exploring the new object (Tnew ×100)/(Tf + Tnew), where Tf and Tnew are the times spent exploring the familiar object and the novel object, respectively. The time spent exploring each object was scored by an observer “blind” to genotypes and in between trials. All of the objects were cleaned with 5% ethanol solution after each trial. Exploration was defined as sniffing or touching the object with the nose or forepaws. 

### 2.3. Contextual Fear Conditioning (CFC) and Tone Fear Recall

The procedure was adapted from [24,33]. The setup was an experimental chamber 22 cm × 22 cm × 35 cm (L × W × H) located inside a larger noise-attenuating box, and a built-in ventilation fan provided a background noise. The floor of the box was made of stainless-steel rods designed for mice and connected to a scrambled shock generator (TSE Systems) containing a speaker for audible tone. The CFC experiment took place on seven consecutive days. On day 0, the mice were allowed to freely explore the conditioning chamber for 3 min and baseline freezing was measured. Immediately after that, conditioned stimuli tone (75 dB, 2 kHz, 30 s) paired with unconditioned stimuli foot shock (1 s, 0.50 mA, constant electric current) was automatically delivered with 1 min intervals for three times through a grid floor. After completing the conditioning session, the mice were returned to their home cage. On day 1, contextual fear retention was assessed at 24 h after the conditioning session by placing animals into conditioned context for 3 min in the absence of tone and foot shock, during which the duration of freezing time (absence of any movement other than that due to respiration) was measured. The extinction of contextual fear memory was measured from day 2 to day 6. Each day, animals were placed in the same context for 3 min and the freezing time was recorded. On day 7, tone fear recall was assessed by placing the mice in the novel context for 3 min and baseline freezing was measured. Immediately after that, tone (75 dB, 2 kHz, 30 s) was presented and freezing time was measured within the next three minutes.

### 2.4. Open Field Test (OFT)

OFT was used to assess anxiety and locomotion was performed according to [34]. Mice were placed in an experimental room for about 1 h before starting the experiment for their habituation. Each mouse was placed in the center of OFT chamber (45 cm × 45 cm × 45 cm) and allowed to explore freely for 30 min. The light luminosity was set to 500 lux throughout the box. During this time, the mice were monitored and data were collected and recorded by an analytical system (TSE Systems, Chesterfield, VA, USA). Anxiety was quantified by measuring the time spent by mice in central sector of the open field, while locomotor activity was measured by estimating total distance travelled.

### 2.5. Elevated Plus Maze (EPM)

The EPM test measuring anxiety was performed according to [32]; it was carried out in a plus-maze with setup consisting of a central zone (5 cm × 5 cm), two open arms (45 cm × 10 cm) and two arms closed by walls (45 cm × 10 cm and 15 cm in height). The maze is elevated 60 cm above the floor level. The animals were placed on the central zone of the EPM and allowed to explore plus-maze freely for 5 min with light luminosity set to 40 lux. A live video-tracking system (Noldus, with EthoVision XT version 8 software, Wageningen, the Netherlands) was used for automated animal tracking and data collection to measure the total number of entries, number of entries onto the open arms and time spent on the open arms. An observer “blind” to genotypes also scored the behavior. The level of anxiety was calculated as percentage of entries onto open arms and percentage of time spent on the open arms.

### 2.6. Tail Suspension Test (TST)

TST was performed as previously described by [35]. This test was used to measure learned hopelessness in mice, where the animals are placed to an inescapable stressful situation by hanging them in separate sections of the test apparatus on a wooden bar by the tip of their tail using an adhesive tape. During the 6 min test period, the behavior was recorded with a camera and the duration of immobility during the testing period was measured. Immobility was defined as a complete lack of movement other than respiration. However, small movements of forefeet and swinging caused by earlier movements were also scored as immobility. An observer “blind” to genotypes scored behavior.

### 2.7. Social Dominance Test (SDT)

SDT was adapted from [36]. This test was performed by placing the mice of different genotypes simultaneously into the opposite ends of a transparent plastic tube (30 cm long, 4.0 cm inner diameter). When the animals interacted in the tube, the more dominant animal forced its opponent out of the tube. The animal with four paws out of the tube was declared as loser, while the animal remaining inside the tube was considered the winner. Each match was set within 2 min. Matches lasting > 2 min were scored as “even”. Animal pairs were decided according to their matched body weight and each animal was encountered with the opponent for three rounds. Numbers of losses, wins and evens were counted and an average of three rounds was taken as percentage of win to assess the social dominance.

### 2.8. Flow Cytometry

Mice were euthanized with CO2. Dissected brain tissues (hippocampus and frontal cortex) were mechanically dissociated through 70 µm cell strainers (352350, BD Bioscience, San Jose, CA, USA) in ice-cold flow buffer (phosphate buffered saline (PBS) with 1% fetal calf serum). Isolated cells were then blocked with 10% rat serum in ice-cold flow buffer for 1 h at 4 °C. The cells were stained with 0.5 µL of the following antibodies: anti-mouse CD11b-BV421 (101251, Biolegend, San Diego, CA, USA), CD45-Brilliant Violet 650 (103151, Biolegend, San Diego, CA, USA), MHCII-Brilliant Violet 711 (cat no. 107643, Biolegend, San Diego, CA, USA), GLAST-APC (130-123-555, Miltenyi, Bergisch Gladbach, Germany) and O4-PE (130-117-357, Miltenyi, Bergisch Gladbach, Germany) with the corresponding isotype control antibodies rat IgG2b-BV421 (400639), rat IgG2b-BV650 (400651), rat IgG2b-BV711 (400653), mouse IgG2a-APC (400219) and mouse IgM-PE (401611) (all from Biolegend, San Diego, CA, USA) in flow buffer for 1 h. After staining, cells were fixed by 4% paraformaldehyde (PFA), permeabilized with PBS containing 0.05% TritonX-100 at 4 °C for 30 min and incubated with an anti-VGLUT2 mAb-Alexa488 (MAB5504A4 Millipore, Burlington, MA, USA) or isotype control mAb-Alexa488 (400132 Biolegend) for 1 h at 4 °C. The cells were washed with PBS, resuspended and acquired with Fortessa flow cytometer (BD Bioscience, San Jose, CA, USA). In total, 100,000 events were recorded in all the samples. Data were analyzed by Kaluza v2.1 software (Beckman Coulter, Indianapolis, IN, USA). GLAST was used to measure number of astrocytes among total brain cells, CD11b was used to detect number of microglial cells, O4 was used to measure number of oligodendrocyte precursor cells (OPC), and the negative selection was used to measure the number of neurons among non-astrocytes. Number of VGLUT2+ cells was measured under the neuron gate. For measuring M1 microglial polarization, CD11b and CD45 was used to collect total microglial cells and MHCII was used to measure the percentage of M1 type of microglia.

### 2.9. Brain Volume Assessment and Immunohistochemistry

After behavioral experiments, animals were deeply anesthetized with chloral hydrate (300 mg/kg, i.p.) and transcardially perfused using 0.9% saline and then with 4% PFA in PBS (pH = 7.4). After fixation of the brain in PFA for 24 h, 40 µm-thick sections were cut on a Leica VT1000S vibro-microtome (Leica Microsystems Pvt Ltd., Wetzlar, Germany) and stored at −20 °C in the cryo-protectant (30% ethylene glycol, 30% glycerol in PBS; pH 7.4). 

For measuring the volume of the whole brain and hippocampus, every sixth section was selected and was incubated in a 0.1M TRIS HCl buffer containing 0.025% trypsin and 0.1% CaCl2 for 10 min followed by washing with PBS. The sections were then incubated with Triton X-100 (0.25%) for 1 h and washed with PBS. Hematoxylin solution was first added to the section for about 30 s, followed by incubation with acidic alcohol solution (HCl 1% in ethanol 70%) for 10 s and washing with tap water. Eosin solution was added for 10 s and the sections were placed on the glass with water-based mounting medium (Vector Laboratories, Newark, CA, USA) and cover-slipped. An average of 6–8 sections per animal were analyzed. For the analysis of the volume, sections were scanned using Leica SCN400 scanner (Leica Microsystems Pty Ltd., Wetzlar, Germany). The volumes of the areas of interest were calculated from the surface area, measured by Aperio Imagescope (v12.4.3.5008), and multiplied by the thickness of the sections and distance between sections.

For Ki67, NeuN and Iba1 staining, sections were washed three times in PBS and treated with 2% H2O2 solution for 20 min followed by incubation in 0.01 M citrate buffer (pH 6.0) at 85 °C for 30 min in water bath and then stood for 30 min at room temperature. Sections were then washed two times in PBS and once in PBS containing 0.1% Triton X-100. Blocking was done with solution containing 5% goat serum, 0.5% Tween-20, 0.25% Triton X-100 in 100 mM PBS for 1 h. Ki67 primary antibody (1:200, rabbit monoclonal antibody (SP6), ab16667, Abcam, Cambridge, UK) was added for 24 h, NeuN primary antibody (1:200, rabbit anti-NeuN, D4G40, Cell signaling technology, Danvers, MA, USA) was added for 48 h, and Iba1 primary antibody (1: 700, rabbit anti-Iba1, CAF6806, FUJIFILM Wako Chemicals Europe GmbH, Neuss, Germany) was added for 72 h. All antibodies were added in blocking buffer and incubation was carried out at 4 °C. After being washed three times, the sections were incubated with secondary antibody (1:400 or 1:700, affinity purified goat anti-rabbit biotinylated IgG (H+L), Vector Laboratories) in blocking buffer at room temperature for 1 h. Ki67-, NeuN- and Iba1-positive cells were visualized using peroxydase method (ABC system and diaminobenzidine as chromogen, Vector Laboratories). The sections were dried, cleared with xylol and cover-slipped with mounting medium (Vector Laboratories, Newark, CA, USA). 

For quantifying morphological characteristics of microglia (cell size, cell body size, size dendritic processes), the images from sections stained for Iba1-positive cells were analyzed using image analysis software (ImageJ 1.48v, http://imagej.nih.gov/ij (accessed on 1 June 2022), National Institutes of Health, Rokville Pike, Bethesda, USA). To quantify morphological characteristics of Iba-1-positive cells, cell size, cell body size, size of dendritic processes and cell body size to cell ratio, an algorithm described in [37] was used. Briefly, images were converted into 8-bit format, before “adjusted threshold” and “analyze particles” functions were used to apply intensity thresholds and size filter. To measure the total cell size, the threshold was maintained at the level that was automatically provided by the ImageJ program, and size filter of 150 pixels was applied. To measure the total cell body size, the threshold was lowered 40 points and no size filter was applied.

The counts of Iba-1- or NeuN-positive cells were obtained from images according to the algorithm described previously in [38]. Briefly, images were converted to 8-bit, background was subtracted, and then obtained images were thresholded, binarized and counted using “analyze particles” command in ImageJ software.

### 2.10. Assessment of Neurogenesis in the Adult Mouse Dentate Gyrus

Cell proliferation in the dentate gyrus was assessed using immunohistochemical detection of Ki67 as endogenous marker of proliferating cells as described above. 

Cell survival assessment was performed according to [39]. Briefly, mice received three BrdU injections (100 mg/kg, i.p, Sigma Aldrich, Berlington, MA, USA) in a total dosage of 300 mg/kg separated by the intervals of 2 h. After three weeks of injection, the animals were sacrificed and the brains were sectioned and stored in the cryo-protectant. 

For BrdU immunohistochemistry, sections were incubated in 0.3% H2O2 solution for 30 min, washed three times in PBS and incubated with 0.1M Tris-HCl buffer containing 0.025% trypsin and 0.1% CaCl2 for 10 min followed by 2N HCl solution at 37 °C for 30 min. Blocking solution containing 2% normal goat serum and 0.25% Triton X-100 was added to the sections for 1 h at room temperature. Next, the sections were incubated for 24 h with blocking solution containing rat monoclonal antibody to BrdU (1:300, RF04-2, Bio-Rad) at 4 °C, followed by incubation in biotinylated rabbit anti-rat antibody (1:400, affinity purified, Lot R1121, Vector Laboratories, Newark, CA, USA) for 1 h. BrdU-positive cells were visualized using the peroxidase method (ABC system and diaminobenzidine as chromogen, Vector Laboratories). The sections were dried, cleared with xylol and cover-slipped with mounting medium (Vector Laboratories, Newark, CA, USA). Ki67 and BrdU-positive cells were counted in every sixth section within the dentate gyrus. All counts were performed using an Olympus BX-51 microscope. To estimate the total number of Ki67 and BrdU-positive cells in a given region, every sixth section was analyzed to obtain the sum of cell counts from each animal and then multiplied by six.

For doublecortin immunohistochemistry, sections were washed three times in 0.1 M Tris-Buffered Saline (TBS) and quenched in 3% H2O2 and 10% MetOH solution for 10 min. Sections were washed again in TBS and were blocked in 5% normal goat serum and 0.25% Triton X-100 in TBS for 1 h at room temperature. Next, sections were incubated with primary antibody (1:500, ab18723, Abcam, Cambridge, UK) in blocking solution for 48 h at 4 °C. The sections were rinsed twice with TBS and incubated with secondary antibody biotinylated goat anti-rabbit antibody (1:1000, Ref no BA-1000, Vector Laboratories) diluted in blocking buffer for 2 h. Doublecortin-positive cells were visualized using the standard immunoperoxidase method (ABC system, Vectastain ABC kit PK-6100, Vector Laboratories), with diaminobenzidine (DAB) as the chromogen (DAB Peroxidase Substrate SK-4100, Vector Laboratories). Total number of doublecortin-positive cells in a given region was obtained from every 24th section and the sum of cell counts was acquired and then multiplied by the 24.

For the determination of the phenotype of the newly generated cells, two sections from each animal which survived three weeks after the BrdU injection were analyzed for co-expression of BrdU and neuronal (calbindin, a marker for mature neurons) or glial (glial fibrillary acidic protein, GFAP, a marker for astrocytes) markers. For immunofluorescent double-labelling, sections were incubated with a mixture of anti-BrdU monoclonal antibody (1:200, RF04-2, Bio-Rad, Hercules, CA, USA) and rabbit anti-calbindin antibody (1:800, AB1778, Chemicon International Inc, Temecula, CA, USA) or rabbit anti-GFAP (1:800, Z0334, Dako, Glostrup, Denmark). Secondary antibodies such as goat anti-rat Alexa-594 antibody (1:800, A11007, Invitrogen, Thermo Fisher Scientific, Waltham, MA, USA) or goat anti-rabbit Alexa-488 (1:700, A11034, Invitrogen, Thermo Fisher Scientific) were used. Confocal microscope (LSM 710 Duo, Carl Zeiss Microscopy GmbH, Oberkochen, Germany) equipped with an argon laser was used to visualize fluorescent signals. 3D images were constructed from series of scans taken at 1.5μm intervals from the dentate gyri, using 40× objective and 2× digital zoom. For illustrative images, 100× objectives were used. Data are expressed as a percentage of BrdU-positive cells found in the granule cell layer and hilus of the dentate gyrus that expressed either calbindin or GFAP.

### 2.11. Isolation of Brain Cells

Brain cells were isolated as previously described in [40]. Tissues were mechanically homogenized and passed through a 70 μm nylon cell strainer (352350, BD Bioscience) with approximately 10–15 mL of 1X DPBS supplemented with 0.2% glucose into a 50 mL conical tube. Isotonic Percoll dilutions were made by diluting stock Percoll (GE-healthcare, 17-0891-01, Chicago, IL, USA) at a 9:1 ratio with 10X PBS to make stock isotonic Percoll (SIP), which is considered 100% SIP. The layers of Percoll were created by diluting the 100% SIP with 1X Dulbecco’s Phosphate Buffered Saline (DPBS) to make 70% SIP, 50% SIP and 35% SIP. Obtained homogenate was then centrifuged at 600× *g* for 6 min at room temperature. Supernatant was decanted and the pellet resuspended in 6 mL of 70% SIP. The resuspended homogenate was transferred to a 15 mL tube and 3 mL of 50% SIP was carefully layered over. Another 3 mL of 35% SIP was carefully layered on top of the 50% SIP layer, and 2 mL of 1X DPBS was layered on top of the 35% layer. The prepared 15mL tubes were then centrifuged at 2000× *g* for 20 min at room temperature without brake. Three discrete layers were established after centrifugation. Microglial cells were collected from the interface between 70–50% SIP and astroglial cells were taken from 50–35% SIP; meanwhile, the remaining top layer consisted of myelin, and other cells were collected and subjected to characterization of neuronal marker. All isolated cells were resuspended in sterile 1X DPBS and centrifuged at 600× *g* for 6 min at room temperature to remove any remaining Percoll. Washed cells were subjected to purity check using qPCR and flow cytometry methods. GLAST was used to check purity of astrocytes, while Cx3cr1 and Slc17a6 were used to check purity of microglial and neuronal cells, followed by quantification of miRNA expression using Taqman miRNA assay.

### 2.12. RNA Extraction and RT-qPCR

Total RNAs were extracted from brain tissues (hippocampus) by using TRI Reagent^®^ (TR 118) (Molecular Research Center, Inc., Cincinati, OH, USA). To measure mRNA expression, cDNA was synthesized using RevertAid First Strand cDNA Synthesis Kit (Thermo Fisher Scientific) followed by qPCR using 5 × HOT FIREPol^®^ EvaGreen^®^ qPCR Supermix (Solis BioDyne, Tartu, Estonia) on a QuantStudio 12KFlex instrument (Thermo Fisher Scientific) according to the instructions of the respective manufacturers. Primer sequences for target genes were given in the Appendix A.

Quantification of miRNA expression was carried out using TaqMan^®^ MicroRNA Assays hsa-miR-146a (Assay ID: 000468, Life technologies) and TaqMan^®^ MicroRNA Assays hsa-miR-146b (Assay ID: 001097, Life technologies, Carlsbad, CA, USA) according to the manufacturer’s instructions. For cRNA synthesis, TaqMan^®^ MicroRNA reverse transcription kit (4366596, Thermo Scientific) and for qPCR, 5× HOT FIREPol^®^ Probe qPCR Mix Plus (ROX) (Solis BioDyne) were used, respectively. U6 snRNA (Assay ID: 001973, Life Technologies) was used for the normalization of RT-qPCR. To measure cell-specific expression of miR-146a and miR-146b, the respective cell populations were isolated as described above and then miRNA expression in each cellular population was measured using Taqman miRNA assay.

### 2.13. Statistical Analysis

GraphPad 8.0.1 was used for statistical analyses and graphical presentations. Student’s *t*-test, one-way and two-way ANOVA with Tukey’s test were used for post hoc multiple comparisons for statistical analyses, and statistical significance was set at *p* < 0.05. In all figures, data are shown as mean ± standard error of the mean (SEM). For target search, we used TargetScanHuman Release 8.0 (https://www.targetscan.org/vert_80/, accessed on 1 June 2022) [41,42]. Of the transcripts, a total of 299 were conserved sites and 126 were poorly conserved sites. Pathway analysis was performed with g:GOSt tool available in g:Profiler platform, which estimates significance of overlap between functional groups and list of studied genes by calculating enrichment *p*-value using Fisher’s one-tailed test [43]. Only significantly (*p* < 0.05) overrepresented pathways associated with neuron development and function are shown. The associations between behavior in NORT and *Gdnf* mRNA expression were analyzed using Pearson’s correlation.

## 3. Results

### 3.1. miR-146b Is Highly Expressed in Neuronal Cells in the Mouse Brain

First, we evaluated relative expression of miR-146a/b in the hippocampus of *WT* mice and found that both miR-146a and miR-146b were expressed at similar levels in the hippocampal region of mice brains (*p* = 0.8472; Student’s *t*-test; Figure 1A). Next, to measure expression of miR-146a/b in different cell types, we isolated three cell populations (microglia, astroglia and remaining cell fraction, i.e., other cells) from adult WT brains. As expected, RT-qPCR and flow cytometry analysis of purified cell types showed that microglial marker *Cx3cr1* mRNA was enriched in microglial fraction (F (2, 6) = 10.83, *p* = 0.0102; one-way ANOVA; Figure 1B). Staining with astroglial marker GLAST by flow cytometry showed enriched number of GLAST+ cells in astroglial fraction (F (2, 6) = 70.69, *p* = < 0.0001; one-way ANOVA; Figure 1C). The other cell fraction defined as neuronal cells expressed high levels of neuronal *Slc17a6* mRNA (F (2, 6) = 65711, *p* = <0.0001; one-way ANOVA; Figure 1D). Interestingly, miR-146a was highly expressed in microglial cells and less in astroglial cells, and was much lower in the fraction containing mainly neuronal cells (F (2, 6) = 23.51, *p* = 0.0014; one-way ANOVA; Figure 1E). In contrast, miR-146b was more highly expressed in the neuronal fraction as compared to microglial and astroglial fractions (F (2, 6) = 18.48, *p* = 0.0027; one-way ANOVA; Figure 1F). 

In general, *Mir146b*-/- mice did not present any differences regarding their development, body weight, food and water consumption, and premature mortalities. In addition, no visible abnormalities in brain structures were seen in adult *Mir146b*-/- mice. Volumetric assessment of whole brain (*p* = 0.5936; Student’s *t*-test; Appendix A) and hippocampus (*p* = 0.1356; Student’s *t*-test; Appendix A) did not reveal any changes as compared with *WT* mice. Together, these data demonstrate that despite miR-146b being highly expressed in neurons, lack of miR-146b does not cause major phenotypic impairments. 

### 3.2. Mir146b-/- Mice Display Improved Recognition and Associative Memory

Several recent studies indicate that miR-146a might influence cognitive functions of mice [44,45,46]; however, no such studies had been performed for miR-146b. Therefore, and because miR-146b expression was higher in neuronal brain cells, we next performed a series of experiments to assess cognitive abilities and behavior of miR-146b-deficient mice. First, we performed NORT test, which takes advantage of the natural preference of mice for novel objects and is widely used to evaluate cognition, and in particular, recognition memory [47]. In this test, during the training session there were no significant differences in the time spent by mice of both genotypes exploring the two familiar objects (Appendix A), indicating that both genotypes *Mir146b*-/- and *WT* mice have the same motivation to explore new objects. However, during the test phase, when one of the familiar objects was replaced by another novel object, *Mir146b*-/- mice showed a significant preference for the novel object as compared to *WT* mice, both at 2-h (*p* = 0.0027; Student’s *t*-test; Figure 2A) and 24-h (*p* = <0.0001; Student’s *t*-test; Figure 2B) time-points following the training session. 

Next, we assessed the associative memory of *Mir146b*-/- and *WT* mice by measuring contextual fear retention and extinction and tone fear recall. There were no differences detected in baseline freezing time between *WT* and *Mir146b*-/- mice (Appendix A). The mice were next exposed to the tone paired with foot-shock, and after 24 h of exposure, fear retention was measured in the same context. As expected, both *WT* and *Mir146b*-/- mice showed robust freezing response; however, *Mir146b*-/- mice demonstrated significantly longer freezing time compared to *WT* mice (*p* = 0.0171; Student’s *t*-test; Figure 2C). Fear extinction was assessed in absence of tone and foot shock from day 2 to day 6 in the same context; however, no differences in genotypes were detected between *WT* and *Mir146b*-/- mice (F (1, 95) = 3.117, *p* = 0.0807; two-way ANOVA; Figure 2D). Nevertheless, while tone fear recall was measured on day 7, *Mir146b*-/- mice demonstrated significantly longer freezing time compared to the controls (*p* = 0.0063; Student’s *t*-test; Figure 2E). These results indicate that miR-146b deficient mice have better ability to recognize novel objects and have better fear memory acquisition and recall as compared to *WT* mice.

### 3.3. Mir146b-/- Mice Showed Anxiety-like Behaviors but Had No Differences in Depression-like and Social-Dominant Behaviors

As anxiety is often associated with changes in the cognitive domain of the brain [48], we next evaluated anxiety-like behaviors in *Mir146b*-/- mice using OFT and EPM tests. In OFT, *Mir-146b*-/- mice showed no difference in the total distance travelled in the open field arena (*p* = 0.9592; Student’s *t*-test; Figure 3A) compared to their *WT* controls. However, the time spent in the central area of the open field was significantly lower in *Mir146b*-/- mice, indicating that these mice are more anxious than control animals (*p* = 0.0002; Student’s *t*-test; Figure 3B). However, no changes in the number of total entries (*p* = 0.4684; Student’s *t*-test; Figure 3C) and percentage entries onto open arms of the plus maze (*p* = 0.4470; Student’s *t*-test; Figure 3D) in *Mir146b*-/- mice were observed. In line with OFT test results, *Mir146b*-/- mice showed significant decrease in the percentage of time spent on the open arms of EPM as compared to the *WT* mice (*p* = 0.0027; Student’s *t*-test; Figure 3E). 

Previously, it has been reported that the expression levels of miR-146a/b are inversely correlated with severity of depression in patients with major depressive disorder [49,50]. Therefore, and because the depression is often associated with changes in anxiety and cognitive domain [51], we next evaluated whether miR-146b deficiency may contribute to the depression-like and social dominant behavior using TST and SDT tests. No significant differences between *Mir146b*-/- and *WT* mice were found in TST (*p* = 0.2243; Student’s *t*-test; Appendix A) or SDT tests (*p* = 0.2476; Student’s *t*-test; Appendix A). Together, these results indicate that loss of miR-146b causes anxiety-like behaviors, but does not induce depression-like behaviors or affect social behavior in mice.

### 3.4. Brain Cell Abundancy Is Altered in the Brain of Mir146b-/- Mice

Cellular composition is considered as another informative characteristic to understand the brain functions [52]. Thus, we next used flow cytometry to quantify major cell types, such as astrocytes, microglia, oligodendrocyte precursor cells (OPC) and neuronal cells in hippocampus (HP) and frontal cortex (FC) of *Mir146b*-/- and *WT* mice. 

The gating strategy of brain cells is depicted in the representative dot plots (Figure 4A). Isotype antibody staining as negative controls is shown in (Appendix A). Analysis of the hippocampal cells of *Mir146b*-/- mice demonstrated significantly reduced numbers of astrocytes (*p* = 0.0010; Student’s *t*-test; Figure 4B), and no differences were found in the number of microglia (*p* = 0.6168; Student’s *t*-test; Figure 4C) and OPCs (*p* = 0.0658; Student’s *t*-test; Figure 4D) compared to *WT* mice. Meanwhile, with negative selection we also observed an increased number of neuronal cells (*p* = 0.0033; Student’s *t*-test; Figure 4E), while with VGLUT2 staining we found increased number of VGLUT2+ glutamatergic neurons (*p* = 0.0019; Student’s *t*-test; Figure 4F) in *Mir146b*-/- mice. The changes in cellular content in FC of *Mir146b*-/- mice were similar to HP, showing decreased numbers of astroglia (*p* = 0.0211; Student’s *t*-test; Figure 4G) and no changes in microglial (*p* = 0.6372; Student’s *t*-test; Figure 4H) and OPCs number (*p* = 0.1786; Student’s *t*-test; Figure 4I), whereas increased numbers of neurons (*p* = 0.0339; Student’s *t*-test; Figure 4J) and VGLUT2+ glutamatergic neurons (*p* = < 0.0001; Student’s *t*-test; Figure 4K) were observed. To confirm flow cytometry results, we performed immunohistochemistry analysis using neuronal-specific marker NeuN followed by cell counting using tissue sections from FC. Similarly, with flow cytometry results, we detected increased neuronal density of NeuN+ cells (*p* = 0.0084; Student’s *t*-test; Figure 4L,M) in FC of *Mir146b*-/- as compared to *WT* mice. Altogether, these results suggest that miR-146b deficiency affects the abundance of astroglial and neuronal cells, with no significant changes of microglial and OPC cells in HP and FC.

### 3.5. Increased Hippocampal Neurogenesis in Adult Mir146b-/- Mice

miR-146a has been shown to modulate the cell proliferation and differentiation of various cell types in vitro [17,23]. Therefore, and because *Mir146b*-/- mice had increased numbers of neuronal cells, we next assessed adult hippocampal neurogenesis in miR-146b-deficient mice. For the analysis of proliferative activity, we used immunohistochemical detection of proliferation marker Ki67 in the dentate gyrus of the adult hippocampus. We found that the number of Ki67+ cells in the proliferative zone of dentate gyrus was significantly higher in *Mir146b*-/- mice compared to their *WT* littermates (*p* = 0.0075; Student’s *t*-test; Figure 5A,E). To track cell survival, BrdU was administered (i.p) to the *Mir146b*-/- and *WT* mice, and three weeks later the brains were processed for immunohistochemistry to visualize BrdU+ cells in the dentate gyrus. A significantly higher fraction of survived BrdU+ cells were detected in *Mir146b*-/- mice compared to *WT* mice (*p* = 0.0281; Student’s *t*-test; Figure 5B,F). To assess whether loss of miR-146b affects newly generated cells at earlier stages of differentiation, we used doublecortin to label young post-mitotic neurons and found significantly higher number of doublecortin positive cells in the *Mir146b*-/- mice compared to *WT* mice (*p* = 0.0252; Student’s *t*-test; Figure 5C,G).

In order to determine the phenotype of the BrdU+ cells, we performed immunofluorescence analysis with the antibodies against BrdU, neuronal (calbindin) or astroglial (GFAP) markers. We found no differences in percentage of BrdU+ cells that co-localized with astroglial marker, GFAP (*p* = 0.1545; Student’s *t*-test; Figure 5D,H), whereas percentage of BrdU+ cells that co-localized with calbindin was higher in *Mir146b*-/- mice compared to the *WT* mice (*p* = 0.0002; Student’s *t*-test; Figure 5D,I). These results indicate that miR-146b deficiency results in increased proliferation, survival and differentiation of progenitors to neuronal lineage but not astroglial lineage.

### 3.6. miR-146b Deficiency Does Not Cause Microglial Activation in the Hippocampus

As numerous previous studies have demonstrated the immunomodulatory role of miR-146b [25,27], we further explored in more detail microglial morphology, microglial polarization and the expression of *Il1b*, *Tnf*, *Il18* mRNA. No changes were observed in the density of Iba1-positive microglia (*p* = 0.8618; Student’s *t*-test; Figure 6A,B). We analyzed microglial morphological parameters such as microglial cell size (*p* = 0.7216; Student’s *t*-test; Figure 6D), cell body size (*p* = 0.4740; Student’s *t*-test; Figure 6E) and size of dendritic processes (*p* = 0.6132; Student’s *t*-test; Figure 6F), and found no difference between *WT* and *Mir146b*-/- mice. Next, we evaluated the expression of cytokines *Il1b* (*p* = 0.3853; Student’s *t*-test; Figure 6G), *Tnf* (*p* = 0.3853; Student’s *t*-test; Figure 6H), *Il18* (*p* = 0.2043; Student’s *t*-test; Figure 6I) mRNA levels and found no differences between *Mir146b*-/- and the *WT* mice. Similarly, flow cytometry analysis did not reveal changes in the M1 microglial polarization in *Mir146b*-/- mice (*p* = 0.1780; Student’s *t*-test; Figure 6J). It is established that microglia are involved in the regulation of neurogenesis [53] via interaction of microglial fractalkine receptor Cx3cr1 with neuronal Cx3cl1 [54]. We next assessed whether observed enhancement of hippocampal neurogenesis in miR-146b deficient mice could possibly be mediated via fractalkine signaling and measured the expression of *Cx3cr1* mRNA levels in the hippocampus of miR-146b deficient mice and their *WT* littermates. No differences in *Cx3cr1* mRNA expression were observed between the groups (*p* = 0.6114; Student’s *t*-test; Figure 6K). These results indicate that loss of miR-146b neither influences microglial activation nor affects neurogenesis through fractalkine signaling in HP of *Mir146b*-/- mice.

### 3.7. Glial Cell Line-Derived Neurotrophic Factor (Gdnf) mRNA Is Upregulated in the Hippocampus of Mir146b-/- Mice

To explore which putative miR-146b targets may influence cognition, increased neuronal density in the FC and enhanced hippocampal neurogenesis, we performed a TargetScan [41] search in combination with pathway analysis with g:Profiler [43] as well as a literature search for verified targets. We selected 283 genes containing a total of 299 conserved and 126 poorly conserved sites with TargetScanHuman and subjected this list of genes to pathway and gene ontology group analysis. Interestingly, we detected multiple gene ontology groups associated with neuron development and function to be overrepresented among conserved miR-146a/b targets. Among all selected genes shown in (Appendix A), we detected glia-derived neurotrophic factor (*Gdnf*) as a putative target for miR-146a/b. *Gdnf* is known to be expressed in neurons (https://www.brainrnaseq.org/ accessed on 1 June 2022) [55] and previously miR-146a had been shown to be to negatively regulate Gdnf expression [56]. Based on the literature, we also selected for analysis brain-derived neurotrophic factor (*Bdnf*) as a gene. This gene is expressed in astrocytes (https://www.brainrnaseq.org/ accessed on 1 June 2022) [55] and may alter miR-146b expression as Bdnf mutation Val66Met has been associated with altered expression of miR-146b and its downstream targets [57]. In addition, we choose for analysis miR-146a/b target *Irak1*, which is highly expressed in the glial cells (https://www.brainrnaseq.org/ accessed on 1 June 2022) [55]. The expression analysis of the selected targets by RT-qPCR demonstrated that *Gdnf* mRNA expression levels were significantly higher in the hippocampus of *Mir146b*-/- mice compared to *WT* littermates’ levels (*p* = 0.0477; Student’s *t*-test; Figure 7A), whereas there were no changes in *Irak1* levels (*p* = 0.6547; Student’s *t*-test; Figure 7B) and *Bdnf* mRNA levels levels (*p* = 0.9968; Student’s *t*-test; Figure 7C). These data together suggest that miR-146b has capacity to modulate neuronal development due to its influence on expression of *Gdnf* and possible other factors involved in neuronal development. 

### 3.8. Association between Enhanced Cognition in NORT and Gdnf mRNA Expression of Mir146b-/- Mice

To find out whether the enhanced cognition in NORT has any correlation with enhanced expression of *Gdnf* in the hippocampus, we then employed Pearson’s analysis. We found that percentage of preference for the novel object in NORT was positively correlated with the levels of *Gdnf* mRNA expression at 2-h (r = 0.7617, *p* = 0.0466; Figure 8A) and 24-h (r = 0.7722, *p* = 0.0419; Figure 8B) time points in *Mir146b*-/- mice, while no significant correlation was observed in *WT* mice.

## 4. Discussion

The results of this study show that despite sequence similarities between miR-146a/b, their cellular distribution is remarkably different in the mouse brain tissue. While miR-146a is abundantly expressed in the microglial cells, miR-146b was highly expressed in neurons and less in microglia and astroglia. Because the role of the neuronal miR-146b is completely unknown, we next performed a detailed assessment of behavior as well as cellular organization of the brain in *Mir146b*-/- mice.

One of the major findings of our study is that *Mir146b*-/- mice demonstrated enhanced episodic recognition memory. Furthermore, better memory acquisition and recall were shown in the contextual fear conditioning and tone recall tests. Interestingly, *Mir146b-/-* mice also demonstrated slightly increased anxiety in the OFT and EPM tests. In general, anxiety sensitizes sensory cortical systems to innocuous environmental stimuli and might thereby facilitate cognition. Thus, anxiety might play an important adaptive role in the process of cognition [58]. There is a consensus theory that anxiety is associated with better attention control [59], because of improvement in the selectivity of attention and probable better acquisition of the negative emotional stimuli [60]. Thus, the observed enhancement in cognition functions might be at least in part attributed to the increased anxiety levels and arousal. Despite the observed increased anxiety, the immobility time in TST was not changed, suggesting that miR-146b deficient mice do not demonstrate depression-like behavior. 

Although we did not observe any changes in the volume of the brain and hippocampus, the experiments using flow cytometry showed abnormalities in the cellular composition in the brain of *Mir146b*-/- mice. We found that loss of miR-146b led to the increased number of neurons, decreased astrocytes and increased VGLUT2+ glutamatergic neurons in HP and FC of *Mir146b*-/- mice. Further immunohistochemistry revealed increased density of the neurons in FC sections of miR-146b deficient mice. Therefore, it might be proposed that the miR-146b deficiency results in the loss of control in the growth of the neuronal population, which results in the higher numbers of neuronal cells. In line with our results, it has previously been shown that miR-146b overexpression by lentivirus vector could inhibit the proliferation of primary hippocampal neural stem cells after transfection [26]. Since the neuronal proliferation in the FC is restricted to the prenatal period [61,62], we speculated that miR-146b has influence on neuronal generation during the early stages of the brain development; however, further studies are needed to confirm this hypothesis. In addition, further experiments are needed to explore whether the observed decrease in the astroglia is also developmentally regulated. During development, the cross-regulatory interactions between elements of different pathways affect the process of cell fate assignment during neural and astroglial tissue patterning [63]. Both neurons and astroglia are produced from the radial glial progenitor cells and cross talk between important signaling pathways such as JAK-STAT signaling, Wnt signaling responsible for this switch from neurogenesis to gliogenesis phase and for differentiation of astrocytes [64]. As miR-146b has been shown to regulate the JAK-STAT [16] and Wnt signaling [65], the absence of miR-146b might cause perturbation in Wnt signaling, which in turn results in the reduction in astroglia during the early stages of development. 

We also speculated that capacity of miR-146b to affect neuronal development might persist in the neurogenic region of the adult dentate gyrus. Indeed, mice deficient in miR-146b had enhanced neurogenesis in dentate gyrus, as demonstrated by the higher numbers of neuronal precursors, proliferative cells and their better survival. When assessing the phenotype of newly generated cells surviving, we found that a larger proportion expressed a mature neuronal marker, Calbindin, in miR-146b deficient mice, showing increased differentiation into neurons. As previous studies have demonstrated important roles of the adult dentate gyrus neurogenesis in the memory processing [66,67], we speculate that increased neurogenesis might be responsible for the observed enhanced learning and memory capability in *Mir146b*-/- mice. 

As previously miR-146b had been shown to regulate immune responses in variety of cells and tissues [15,17,27], and because microglia can control adult neurogenesis through secretion of various soluble factors including cytokines [68,69], we studied in more detail the morphology of microglia cells. We did not observe signs of microglial activation in *Mir146b*-/- mice. Similarly, no change was observed in the mRNA levels of *Cx3cr1*, a fractalkine receptor essential in the regulation of the interactions between microglia and neurons [70]. These results suggest that microglia or microglia-derived factors are not involved in the promotion of the neuronal phenotype observed in *Mir146b*-/- mice.

In line with the neuronal phenotype of *Mir146b*-/- mice, the pathway analysis of conserved miR-146b targets revealed that neurogenesis-related genes are overrepresented among miR-146b target genes, indicating that miR-146b might influence neuronal development through several different genes. From putative targets, we selected *Gdnf* and explored its expression in the miR-146b deficient mouse brain. Indeed, we observed an increased expression of *Gdnf* mRNA in the hippocampus of *Mir146b*-/- mice. As *Gdnf* participates in proliferation, migration and differentiation of the neural cells [71], and might direct newly generated neurons to the specific neuronal phenotypes [72], we propose that upregulation of *Gdnf* in *Mir146b*-/- mice might contribute to increased neuronal proliferation and survival these mice. Further *Gdnf* mRNA expression was positively correlated with the observed cognitive behaviors in *Mir146b*-/- mice, thus upregulation of *Gdnf* mRNA expression might contribute to the promotion of enhanced cognition observed in miR-146b deficient mice. As previous studies have shown that *Gdnf* improves spatial learning in aged rats [73,74]. Interestingly, no difference was observed in mRNA levels of *Bdnf*, which is a factor potentially regulating miR-146b levels as well as *Irak1*, which is well-known target of miR-146b [19]. However, additional experiments are needed to assess whether there is effect on additional miR-146b targets. Further research is required to provide insight into causal relationships between behavioral and morphological consequences of miR-146b deletion.

In addition, previous studies have shown that electro acupuncture-stimulated hippocampal neurogenesis in the rat model of focal cerebral ischemia and reperfusion was associated with an increased miR-146b expression, while inhibition of miR-146b reduced stimulatory effect of acupuncture on the hippocampal neurogenesis [75]. It should be noted that, aside from the novel neuronal functions of miR-146b described in our study, miR-146b is expressed in microglial cells and is involved in the negative regulation of neuroinflammation [25,27]. Previous studies have demonstrated that focal cerebral ischemia and reperfusion can induce inflammation and that acupuncture is able to reduce neuroinflammation [76]. It is possible that acupuncture might improve neurogenesis via expression of anti-inflammatory miR-146b and that inhibition of neurogenesis by the miR-146b inhibition might involve an inflammatory component. This possibility should be studied in more detail further.

It is also important that further study is conducted around the possible roles of miR-146b in the regulation of apoptosis. There is compelling evidence that miR-146a is an important negative modulator of apoptosis via TAF9b/P53 [77] and SMAD3 pathways [78]. Recent studies demonstrated that miR-146b is also involved in the regulation of apoptosis and that over-expression of miR-146b promoted cell death [79]. Furthermore, miR-146b-5p inhibits tumorigenesis and metastasis of gall bladder cancer by targeting toll-like receptor 4 via the NF-κB pathway [80]. Thus, it is not ruled out that deficiency in miR-146b will result in the suppression of apoptosis, which might also have an impact on the increased neuronal density in the miR-146b deficient mice. Further research is required to provide insight into possible roles of miR-146b in the regulation of neuronal apoptosis as well as causal relationships between behavioral and morphological consequences of miR-146b deletion.

## 5. Conclusions

In conclusion, our data provide new evidence that miR-146b has important roles in the control of the proliferation and differentiation of neuronal precursors during the development as well as in adulthood. In addition, we show that miR-146b is able to control adult hippocampal neurogenesis, which might be relevant for the observed enhanced cognition and fear. Our data indicate that miR-146b probably exerts its actions via regulation of *Gdnf* expression, which explains why deficiency in miR-146b leads to the activation of neurogenesis. Our data also open new avenues for the regulation of hippocampal neurogenesis via modulation of miR-146b. In addition, miR-146b might be used as a biomarker for tumors of neuronal origin.

## Figures and Tables

**Figure 1 cells-11-02002-f001:**
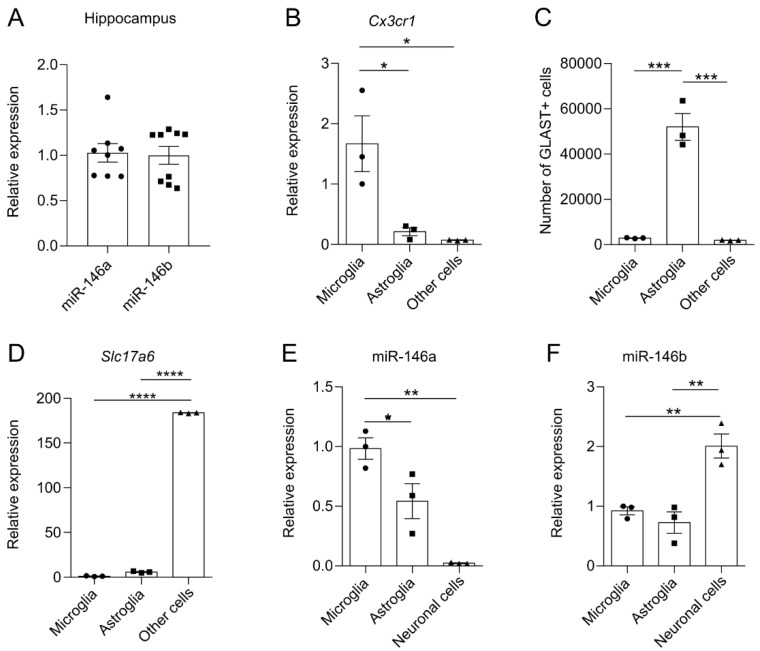
Expression of miR-146a and miR-146b in hippocampus and cellular fractions isolated from WT mice. (**A**) Quantification of miR-146a/b expression in the hippocampus of WT mice. Number of animals in (**A**) = 8–9, Student’s *t*-test. (**B**–**D**) Characteristics of the cellular fractions isolated from the *WT* mouse brain. (**B**) Relative mRNA expression of microglial *Cx3cr1*. (**C**) Number of astroglial marker GLAST+ cells (determined by flow cytometry). (**D**) Relative expression of neuronal marker *Slc17a6* mRNA levels. As fraction of “other cells” abundantly expressed neuronal marker *Slc17a6* mRNA, this fraction was named as neuronal cells. (**E**) Quantification of miR-146a and (**F**) miR-146b in microglia, astroglia and neuronal cells of *WT* mice. Number of cell batches in experiments (**B**–**F**) = 3, one-way ANOVA followed by Tukey’s multiple comparison test. Data represented as mean ± SEM; * *p* < 0.05, ** *p* < 0.01, *** *p* < 0.001, **** *p* < 0.0001.

**Figure 2 cells-11-02002-f002:**
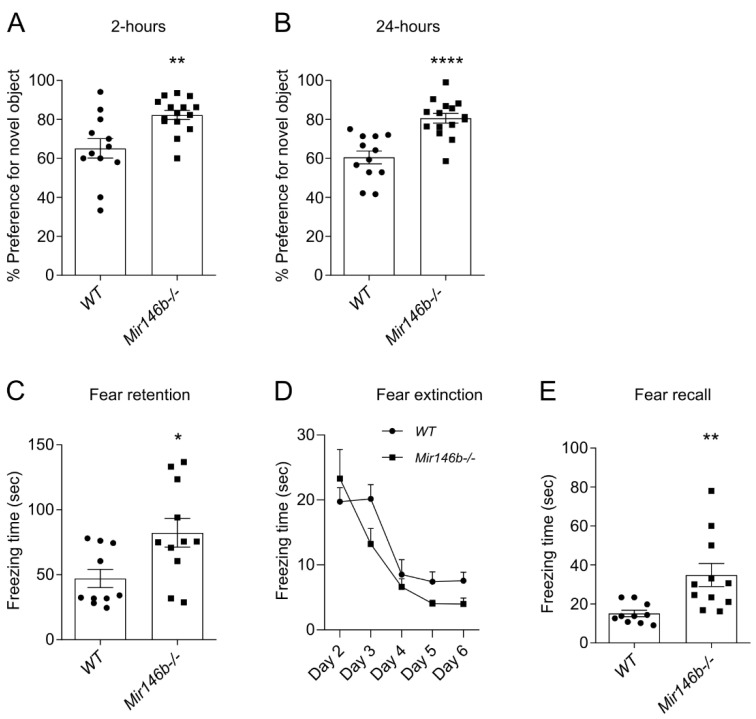
Enhanced recognition and associative memory in *Mir146b*-/- mice. (**A**) Novel object preference of mice at 2-h (short-term memory) and (**B**) 24-h (long-term memory) in novel object recognition test. (**C**) Contextual fear retention. (**D**) Contextual fear extinction and (**E**) tone fear recall of *WT* and *Mir146b*-/- mice in contextual fear conditioning and tone fear recall. Number of animals = 11–15, Student’s *t*-test and two-way ANOVA followed by Tukey’s multiple comparison test. Data represented as mean ± SEM; * *p*  <  0.05, ** *p*  <  0.01, **** *p* <0.0001.

**Figure 3 cells-11-02002-f003:**
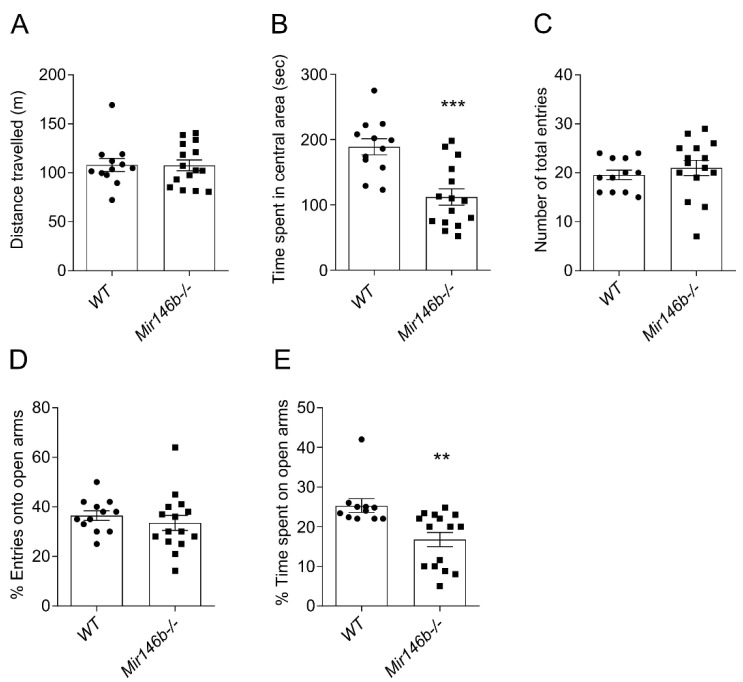
miR-146b deficiency causes anxiety-related behaviors. (**A**) Total distance travelled (locomotor activity). (**B**) Time spent in the central area of the open field test. (**C**) Total number of entries. (**D**) Percentage of entries onto the open arms. (**E**) Percentage of time spent on the open arms of elevated plus maze of *Mir146b*-/- mice and their *WT* littermates. Number of animals = 11–15, Student’s *t*-test. Data represented as mean ± SEM; ** *p* < 0.01, *** *p* < 0.001.

**Figure 4 cells-11-02002-f004:**
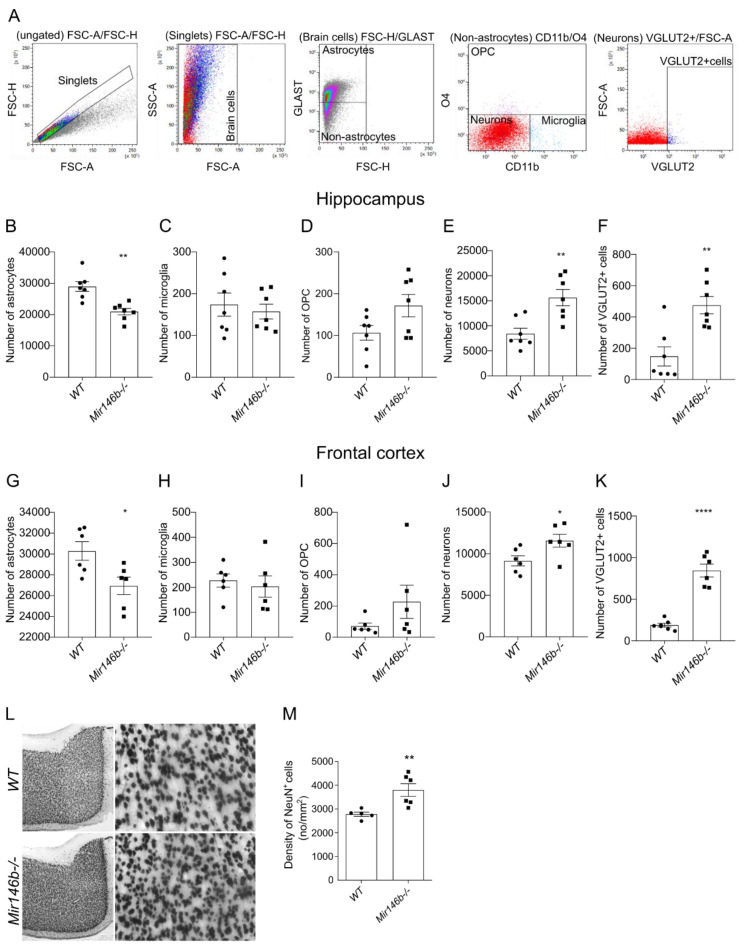
miR-146b deficiency causes altered brain cell abundancy in the hippocampus and frontal cortex of the mouse brain determined by flow cytometry and immunohistochemistry. (**A**) Representative graphs showing flow cytometry gating strategy for astrocytes, microglia, oligodendrocyte precursor cells (OPC) and neurons. (**B**) Number of astrocytes among total brain cells. (**C**) Number of microglia, (**D**) OPC and (**E**) neurons among non-astrocytes. (**F**) Number of VGLUT2+ neurons in the hippocampus and (**G**–**K**) frontal cortex of *WT* and *Mir146b*-/- mice. (**L**) Representative immunohistochemistry microphotographs of the frontal cortex NeuN+ sections at 40× and 200× magnification and (**M**) quantification of NeuN-positive neuronal density in the frontal cortex of *WT* and *Mir146b-/-* mice. Number of animals = 5–7, Student’s *t*-test. Data represented as mean ± SEM; * *p* < 0.05, ** *p* < 0.01, **** *p* < 0.0001.

**Figure 5 cells-11-02002-f005:**
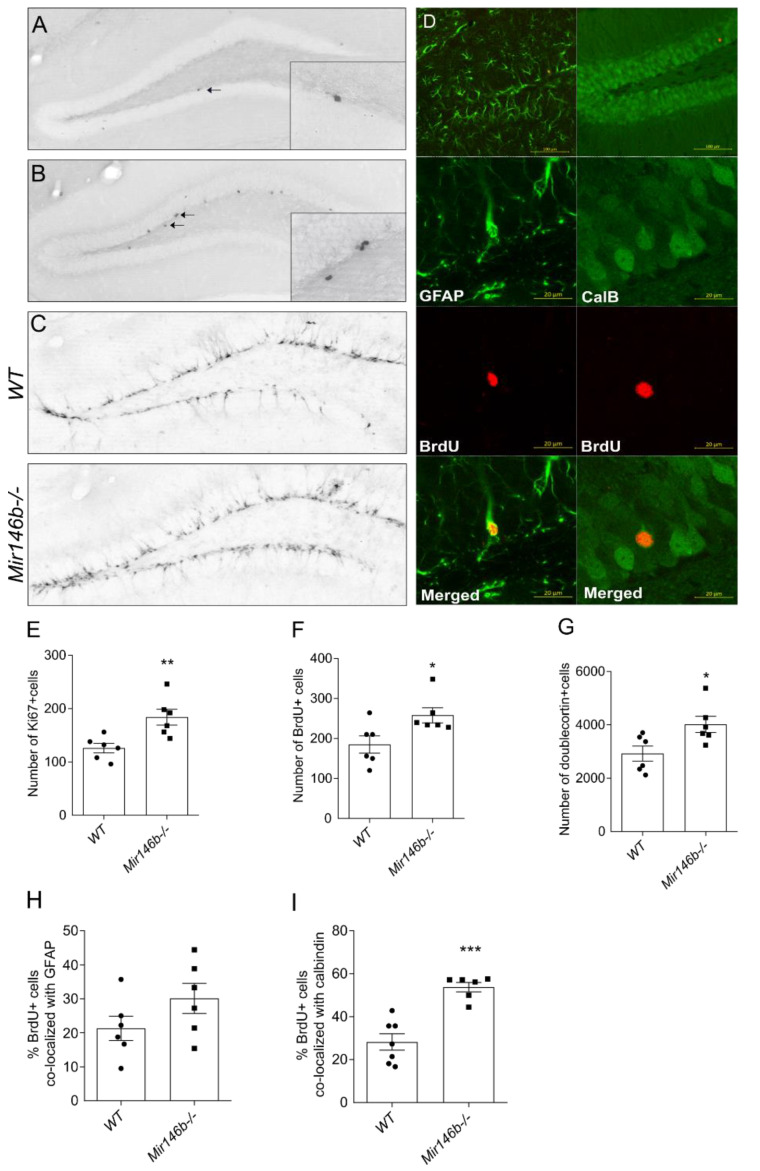
Increased adult hippocampal neurogenesis in *Mir146b*-/- mice. (**A**) Representative immunohistochemistry microphotographs of the hippocampal Ki67+ cells at 100× magnification and inserted microphotographs at 1000× magnification. (**B**) Illustrative microphotographs of the hippocampal BrdU+ cells at 100× magnification and inserted microphotographs at 1000× magnification. (**C**) Represented microphotographs of doublecortin positive cells taken at 100× magnification. (**D**) Illustrative images of BrdU, GFAP and calbindin signal and their co-localization in the hippocampus. (**E**) Quantitative graph showing increased number of Ki67 positive cells. (**F**) Increased number of BrdU+ cells. (**G**) Quantitative graphs showing number of doublecortin positive cells in the hippocampus of *WT* and *Mir146b*-/- mice. (**H**) Percentage of BrdU+ cells with GFAP and (**I**) calbindin in the dentate gyrus of *WT* and *Mir146b*-/- mice. Scale bar = 20 μm. Number of animals = 6, Student’s *t*-test. Data represented as mean ± SEM; * *p* < 0.05, ** *p* < 0.01, *** *p* < 0.001.

**Figure 6 cells-11-02002-f006:**
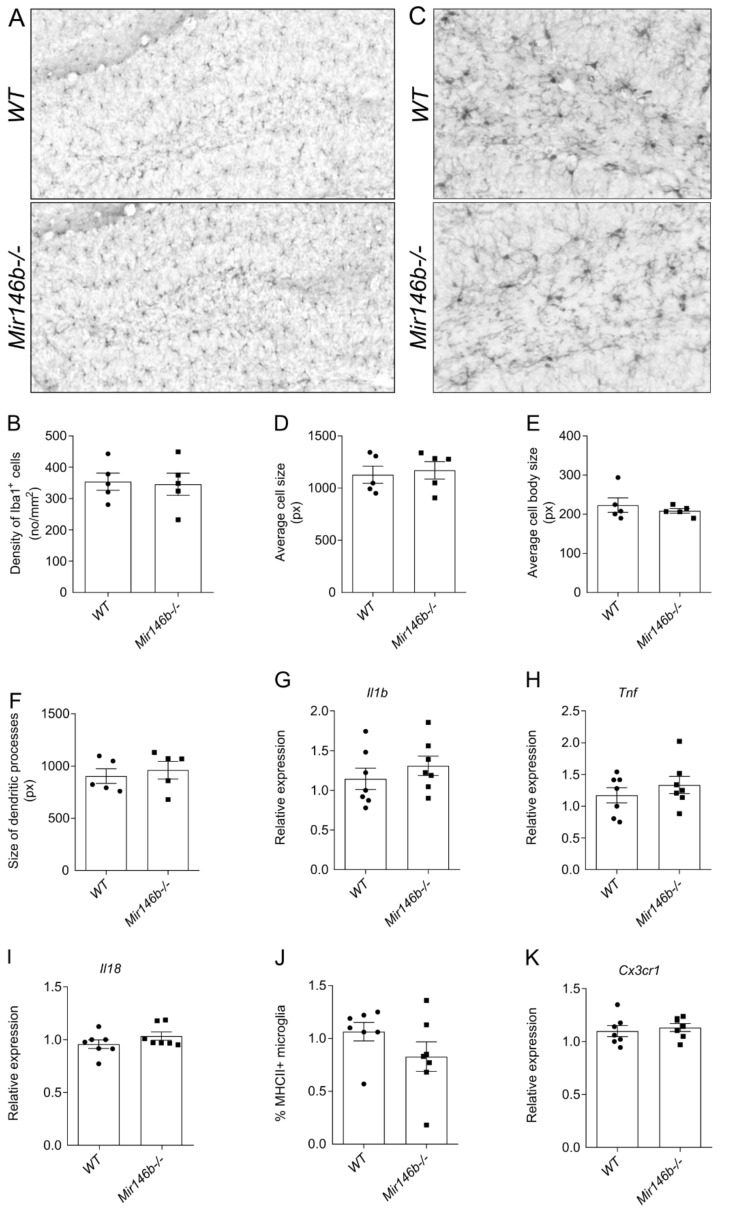
miR-146b deficiency does not cause microglial activation and signs of neuroinflammation in the hippocampus. (**A**) Representative images showing Iba1-positive cells at 100× magnification. (**B**) Quantitative graphs of Iba1 counting in *WT* and *Mir146b*-/- mice. (**C**) Representative images of Iba1 immunohistochemistry for morphological analysis at 400× magnification in the hippocampus of *WT* and *Mir146b*-/- mice. (**D**) Average cell size in pixels. (**E**) Cell body size in pixels. (**F**) Size of dendritic processes in pixels of microglial cells in the hippocampus. Number of animals = 5, Student’s *t*-test. (**G**) Relative mRNA expression of cytokines *IL1b*, (**H**) *Tnf* and (**I**) *IL18.* (**J**) Flow cytometry quantification of percentage of MHCII+ M1 type of microglia. (**K**) Relative mRNA expression of *Cx3cr1* in the hippocampus of *WT* and *Mir146b*-/- mice. Number of animals = 7, Student’s *t*-test. Data represented as mean ± SEM, respectively.

**Figure 7 cells-11-02002-f007:**
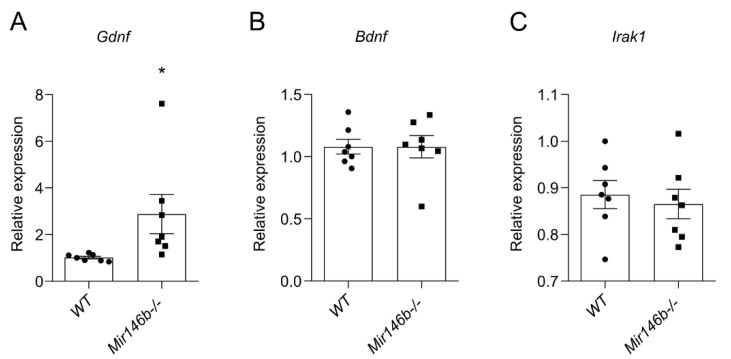
*Gdnf* mRNA is upregulated in *Mir146b*-/- mice. (**A**) Relative mRNA expression of miR-146b targets *Gdnf*. (**B**) *Bdnf* and (**C**) *Irak1* of *WT* and *Mir146b*-/- mice. Number of animals = 7, Student’s *t*-test. Data represented as mean ± SEM respectively; * *p*  <  0.05.

**Figure 8 cells-11-02002-f008:**
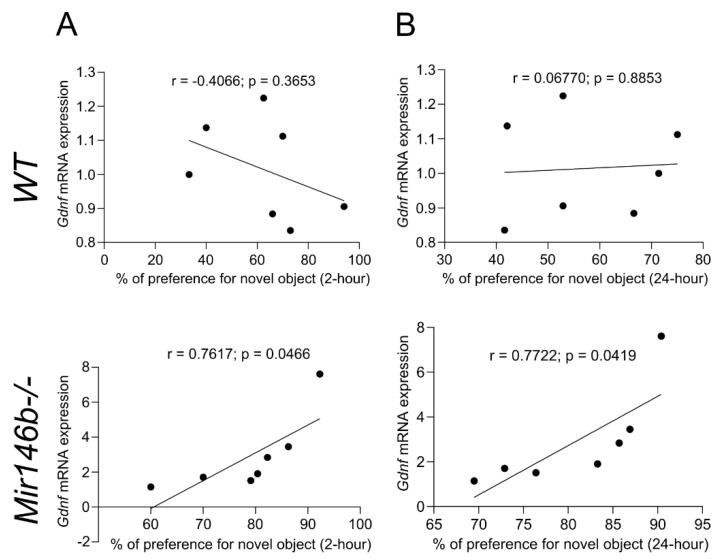
Pearson correlation of percentage of preference for the novel object in NORT with *Gdnf* mRNA expression in the hippocampus of *WT* and miR-146b deficient mice. Correlation analysis was performed at 2 h (**A**) and 24 h (**B**) in NORT.

**Table 1 cells-11-02002-t001:** Sequence of the experiments.

Time Point	Task Assigned
**Cohort 1 (*WT* and *Mir146b*-/-)**	
Day 0	Open field test
Day 4	Elevated plus maze
Day 8–10	Novel object recognition test
Day 13–14	Social dominance test
Day 17	Tail suspension test
Day 22	Sacrifice the animals and collect the tissues for flow cytometry, qPCR and immunohistochemistry
**Cohort 2 (*WT* and *Mir146b*-/-)**	
Day 0–7	Contextual fear conditioning
Day 10	BrdU injections (300 mg/kg)
Day 31	Sacrifice animals for immunohistochemistry and qPCR

## Data Availability

The data presented in this study are available on request from the corresponding author.

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
