# Peer review of "Enhanced Cognition and Neurogenesis in miR-146b Deficient Mice"

_cells, 2022, doi:10.3390/cells11132002_

Round 1

Reviewer 1 Report

Authors investigated the effects of miR-146b deficiency on hippocampal function and neurogenesis using miR-146b knockout mice. Authors used various experimental tools to show the correlation between hippocampal neurogenesis and miR-146b. The results are interesting, but there are some concerns to consider for publication.

Authors should demonstrate the age of animals used in this study. In addition, authors should show the experimental design of the study including sequence of behavioral test and sacrifice time after last experiment. 

Authors showed microphotographs of Ki67, DCX, and BrdU/CalB in Figure 5. Authors should perform immunohistochemical staining for Ki67 again because generally Ki67 positive cells are numerous compared to BrdU positive cells.

Zhang et al (PMID: 32792909) showed that miR-146b inhibitors furtherly suppressed oxygen-glucose deprivation and reperfusion-induced differentiation of neural stem cells, simultaneously NeuroD1 was involved in neural stem cells differentiation into neurons. Authors should cite the reference and discuss the different results of the study.

Reviewer 2 Report

This article is a well-reasoned and well-written work. The abstract section is largely exhaustive and it can be somewhat shortened and generalized, since the data presented make it possible to do so. The Introduction section is written rather briefly, but quite constructively and meaningfully, the rationale for the purpose of the study is confirmed by modern data from other authors. The Material and Methods section is written methodically and completely, which leaves a good impression of the authors' methodological preparedness. The results of the study are well presented, concise and confirmed by statistical analysis. Overall, the results section is quite convincing. The discussion of this work could be expanded, becouse the authors received a rather voluminous factual material. The conclusion also needs to be somewhat expanded and given the practical significance of the results obtained, as well as a conclusion about where and how the data obtained by the authors can be used. In general, the work makes a satisfactory impression and, after a slight revision, can be recommended for publication in the journal Cells.

Reviewer 3 Report

In this study, Chithanathan and collaborators investigated the roles of miR146b on behaviour and neurogenesis in mice. miR146b KO mice display enhanced memory and increased anxiety behaviour, which correlates with an increase in diverse aspects of adult hippocampal neurogenesis. This study is scientifically sound, it was well designed and described. Some points of concern were raised and should be addressed.

1. Why did authors use animals from 8 to 12 weeks? Could this difference in age be a confounding factor for neurogenesis, miR expression or the behavioural testing?

2. Does a wood box (line 107) not retain odour, which can then be used as olfactory cues for mice? Is 5% ethanol sufficient to get rid of these cues?

3. In the open field test, how was the luminosity provided (high luminosity, standard room, red light)? This could affect the interpretation of the results.

4. There is a lost sentence in line 170.

5. From the explanation provided, it is understood that in the flow cytometry experiments, the authors considered neurons everything else that was not positive for the other markers. Is that correct? If so, how can the author exclude the presence of other cell types, e.g. vascular tissue, ependymal cells, etc?

6. Please provide the % for cryo-protectant solution (line 216).

7. In line 238, it is not clear which antibody was left for how long.

8. In line 271, please provide a time range instead of overnight (e.g. 18-24 h).

9. Why DCX cells were counted every 24th slice and not 6 like in the other experiment?

10. Is there a reason why the authors used a Taqman for human samples to analyse miRNA in mouse tissue? Is the miRNA conserved across species?

11. In results 3.1, what about oligodendrocytes?

12. Why n=6 in fig 1A and =3 in fig 1B-F?

13. In line 646, the term "coping with stress" referring to TST could be misleading.

14. The hippocampus can be subdivided along its dorsoventral axis into distinct sub-regions (dorsal, intermediate and ventral). These subregions have been shown to play different roles in learning&memory and anxiety, and the neurogenesis within these subregions is also differentially affected by stress, antidepressants, etc. The authors did not describe which hippocampus slices where analysed and it would be interesting to display the neurogenesis results for each sub-region separately. This is particularly important because more neurogenesis in dorsal hippocampus correlates with memory behaviour, but stress-induced anxiety may reduce neurogenesis in the ventral hippocampus.

Round 2

Reviewer 1 Report

The manuscript has been improved and I have no further comments.